# Impact of Different Frozen Dough Technology on the Quality and Gluten Structure of Steamed Buns

**DOI:** 10.3390/foods11233833

**Published:** 2022-11-27

**Authors:** Bailu Yang, Yining Zhang, Jiayi Yuan, Minzhen Yang, Runqiang Yang, Zhenxin Gu, Chong Xie, Qin Zhou, Dong Jiang, Jianzhong Zhou, Pei Wang

**Affiliations:** 1Whole Grain Food Engineering Research Center, College of Food Science and Technology, Nanjing Agricultural University, Nanjing 210095, China; 2National Technique Innovation Center for Regional Wheat Production, Key Laboratory of Crop Physiology, Ecology and Management, Ministry of Agriculture, National Engineering and Technology Center for Information Agriculture, Nanjing Agricultural University, Nanjing 210095, China; 3College of Food Science and Pharmacy, Xinjiang Agricultural University, Urumqi 830052, China

**Keywords:** frozen dough, steamed buns, processing technology, gluten proteins

## Abstract

To advance the industrialization production of steamed buns, the current study explored the freeze-stability of unfermented, pre-fermented and par-steamed frozen dough. The results showed that the steamed bun made from unfermented dough with 2.0% yeast, the pre-fermented dough with a pre-fermented time of 30 min and the par-steamed dough with a pre-steamed time of 15 min showed the best sensory properties quality upon frozen storage. The gassing power of un- and pre-fermented dough gradually decreased, and dough with longer pre-fermented time exhibited more evident loss of gassing power. Freeze-induced depolymerization of gluten protein was the least distinct in the par-steamed dough, followed by the pre- and un-fermented dough, which was probably related to the superior freeze stability of glutenin-gliadin macro-crosslinks upon the pre-steaming stage. The surface hydrophobicity of gluten proteins of frozen dough decreased during the initial storage and was enhanced subsequently, which was related with the combined effects of the unfolding and synchronous aggregation induced by freezing and steaming, respectively. Moreover, the surface hydrophobicity of gluten in par-steamed frozen dough and steamed buns was more resistant to frozen storage, which was probably attributed to the established stable structure during the pre-steaming process.

## 1. Introduction

The Chinese steamed bun, also known as “Baozi”, is a traditional staple food stemming from China and is widely consumed in Asian countries. Unlike steamed bread without fillings, a steamed bun is usually filled with sweet or savory stuffing, providing diverse tastes. Sweet stuffings typically contain red bean paste or custard, whereas savory stuffings are usually a mixture of meat and vegetables [1]. In China, most steamed buns are freshly produced on a limited scale and consumed immediately in restaurants and breakfast stores. The industrialization of the manufacturing process yields pivotal savings in time and cost. However, the short shelf-life is the predominant obstacle for the industrialization production of steamed buns. Incorporation of additives, vacuum packaging and freezing techniques have been widely studied to prolong the shelf life. Although freezing is a safe and universal technique to preserve foods, frozen steamed buns have not been widely accepted by consumers as a consequence of significant degraded sensory properties during storage, such as enhanced hardness, flavor loss, etc. [2].

Frozen dough techniques have been successfully utilized in Western baked bread. They not only ensure the freshness of baked products, but save costs and insure the standardization of product quality [3]. Unfermented, pre-fermented and par-baked frozen dough involving varied processing techniques are the major products [4]. However, the distinct drawbacks are that freezing and frozen storage can reduce loaf volume and enhance the firmness of bread, which are mainly attributed to water sublimation, a disrupted gluten network and loss of yeast activity [5].

By introducing the freezing technique, the industrialization of steamed buns can be realized. A majority of research has been focused on unfermented frozen dough, whereas the technique of pre-fermented and par-steamed dough remains largely unexplored in steamed buns. Wang et al. evaluated the repeated freeze-thaw treatment of the processing quality of steamed buns with a pre-fermented time of 30 min, and suggested that the deteriorated gluten network and loss of yeast activity were the main reasons for the degraded quality [6]. However, key processing parameters such as the pre-fermented time have not been investigated. Upon the steaming stage, the heat-induced polymerization of glutenin and gliadin via the disulfide (SS) bridge directly affects the loaf volume and texture of the baked bread [7,8]. Previous studies have indicated that the suppressed polymerization of glutenin and gliadin in un-fermented frozen dough further resulted in the increased firmness of steamed breads [9]. However, the dynamic variation of gluten proteins in the pre-fermented and par-steamed dough upon frozen storage remains largely unknown.

Considering this background, the aim of current study is to explore the application of different frozen dough techniques on preserving the steamed buns’ quality. The gassing power and gluten structure in the dough and bun were comparatively studied to depict the attributes for the deteriorated quality of steamed buns upon storage. The results of this study could contribute to improving frozen dough technology as well as the advancing industrialization production of steamed buns.

## 2. Materials and Methods

### 2.1. Materials

Wheat flour (11.6% protein, 14% moisture, 75.6% carbohydrate) for making steamed buns was purchased from Jiangsu Taixing Quxia Flour Co., Ltd. (Taixing, China). Dried yeast (Angel yeast Co., Ltd., Yichang, China) and sugar (Lvzi Food Co., Ltd., Shouguang, China) were purchased from a local supermarket, and red bean paste filling was purchased from Zhanyi baking Co., Ltd. (Shanghai, China). All the reagents used were of analytical grade unless otherwise specified.

### 2.2. Preparation of Frozen Dough and Steamed-Bun-Making Procedure

Steamed buns were prepared according to Zhao et al. [5] with modifications. The basic recipe contained 300 g flour, 156 g water, 3 g yeast and 3 g sugar. The ingredients were homogeneously mixed in a dough mixer (C-100 Mixer, Hobart Corporation, Troy, OH, USA) at 60 and 120 rpm for 2 and 3 min, respectively. Then the dough was divided into 60 g pieces, molded and filled with 24 g of red bean paste.

For the unfermented frozen dough, the yeast level was set at 0.5%, 1% and 2% (*w*/*w* flour basis), and the bun dough was immediately sealed with a plastic membrane, frozen at −40 °C for 12 h and further stored at −18 °C. The unfrozen dough was set as the control group. A batch of dough was freeze-dried after the fixed storage time and the other batch was thawed at 4 °C for 8 h and fermented at 30 ± 2 °C under 80 ± 5% relative humidity to achieve the optimum height. Then, the dough was steamed in the tray above boiling water for 20 min, cooled for 2 h, packed into plastic bags and analyzed within 12 h.

For the pre-fermented and par-steamed frozen dough, 1% yeast was used in the basic recipe. The pre-fermented time was set at 20, 30 and 40 min, respectively. For the par-steamed frozen dough, the par-steamed time was 10, 15 and 20 min. The freezing, thawing and further steamed-bun-making procedure was the same as the above except that the fermented and steamed time were adjusted to achieve the optimum height and completely steamed for the pre-fermented and par-steamed frozen dough, respectively. The unfrozen pre-fermented and par-steamed doughs were designated as the corresponding control groups.

### 2.3. Quality Evaluation of Steamed Buns

The rapeseed replacement method was used to evaluate the specific volume of the steamed buns. The texture profile analysis of the steamed buns was conducted on a TA.XT2i texture analyzer (Stable Micro Systems, Ltd., Godalming, UK) with a P/36R probe [10]. The steamed bun was placed horizontally on the load-bearing platform, and compressed at a speed of 1.0 mm/s to compress the buns to 30%. The moisture of the skin, crumb and filling of the steamed buns (2 g) was analyzed by an Ohaus Halogen moisture analyzer (Ohaus, Switzerland) [11]. Images of the buns were captured using a HP Scanjet 5100C Photo Scanner (Hewlett-Packard, Palo Alto, CA, USA) and the crumb grain structure was analyzed by Image J software v. 1.49 (NIH, Bethesda, MD, USA) [12]. The sensory evaluation of the steamed buns was carried out within 5 h of steamed bun making. Samples were given to thirty trained panelists (15 females and 15 males, age range 20–30) who are asked to evaluate the color, tissue, texture, taste, odor, chewiness and stickiness. Sensory evaluation informed consent was obtained from each subject prior to their participation in the study. The scoring method for the sensory evaluation of steamed busn was evaluated according to the Chinese standard method GB/T 17320-2013 [13] and Zhao et al. [14] with some modifications, as indicated in Table 1. Characteristics were assessed as follows: color (15 points), tissue (15 points), texture (15 points), taste (15 points), odor (10 points), chewiness (15 points) and stickiness (15 points), with a total score of 100 points.

### 2.4. Gassing Power Analysis

Thawed dough (80 g) was placed in a 1000 mL vessel with an open valve, and then warmed in a water bath at 30 °C. When the dough temperature reached 30 °C, the valve was closed. The vessel was linked to an inverted test tube filled with distilled water of pH 2, which was then hermetically closed. The gassing power was reflected by the CO_2_ volume, which was measured through the displacement of water in the test tube with a testing time of 120 min [15].

### 2.5. Free Sulfhydryl (SH) Content

The lyophilized sample (100 mg) was twice shaken with 1 mL of deionized water and centrifuged at 8000× *g* for 5 min, and the pellet (50 mg) was extracted with 2 mL of Tris-glycine-EDTA buffer (TGE, 86 mM Tris-HCl, 4.1 mM EDTA, 92 mM glycine, pH 8.0) in the presence of 2.5% (*v*/*v*) SDS for 30 min, and 10 μL of Ellman’s reagent (DTNB was dissolved in 4 mg/mL TGE) and reacted in the dark for 30 min. The supernatant was measured at 412 nm against the blank and the absorbance was converted to amount of free SH using a calibration curve with reduced glutathione ranging from 0 to 0.1 mM [16].

### 2.6. Molecular Weight Distribution of Gluten Protein

The molecular weight (Mw) distribution of the gluten protein was determined by an Agilent 1200 HPLC system (Agilent Technologies, SantaClara, CA, USA) [17]. Lyophilized samples (10 mg) were shaken with 5 mL of sodium phosphate buffer (PBS, 0.05 M, pH 6.8) containing 2.0% SDS for 1 h at room temperature [18]. After centrifugation at 10,000× *g* for 5 min, the supernatant (20 μL) was loaded on a Shodex Protein KW-804 column (Showa, Kyoto, Japan). The eluent was PBS (0.05 M, pH 6.8) with 0.2% (*w*/*v*) SDS, and separation was conducted at 30 °C with a flow rate of 0.7 mL/min. The detection wavelength was set at 214 nm. SDS-soluble polymers (SDS-P), monomers (SDS-M) and insoluble proteins (SDS-I) were calculated from the corresponding peak area and expressed as a percentage of the peak area of reduced gluten dissolved with SDS solution containing 1.0% dithiothreitol (DTT).

### 2.7. Surface Hydrophobicity Analysis

The lyophilized sample (5 mg) was mixed with 1 mL of 50 mM acetic acid solution for 1 h, and centrifuged at 10,000× *g* for 15 min. The supernatant was diluted to varied concentrations, and 10 μL of 8 mM 8-aniline-1-naphthalenesulfonic acid (ANS) solution (dissolved in 0.1 M PBS, pH 5.8) was mixed with 2 mL of the sample solution. The fluorescence intensity was measured by an F-7000 fluorescence spectrometer (Hitachi, Japan) with the excitation and emission wavelength set at 390 and 470 nm, respectively, and the slit width set at 5 nm [19].

### 2.8. Statistics Analysis

All the data were expressed as mean ± standard deviation (SD) of three replicates. One-way analysis of variance (ANOVA) was used to analyze the data, and Duncan analysis in SPSS software (version 13.0 for Windows, SPSS Inc., Chicago, IL, USA) was used to test the data for significance. The probability value of *p* < 0.05 was considered significant.

## 3. Results and Discussion

### 3.1. Effects of Different Processes on the Quality of Frozen Buns

#### 3.1.1. Specific Volume

Specific volume is one of the most basic and important indicators for evaluating the quality of steamed buns. During freezing and frozen storage, the specific volume of steamed buns produced by the three processes decreased to varied degrees during the frozen storage period. As shown in Figure 1A, the specific volume of freshly steamed buns was independent of the yeast dosage. However, frozen dough with 2.0% yeast possessed the highest specific volume during the entire frozen storage period, and the most evident superior effect was noticed from the 30 to 90 d storage. The minimized gap among the 120 d frozen-stored dough with varied yeast levels was probably due to the predominant role of the distorted gluten network in determining the specific volume. For the pre-fermented frozen dough, the specific volume of the steamed buns was dependent on the frozen storage and pre-fermented time (Figure 1B). Steamed buns with the pre-fermented time of 30 min showed the largest specific volume, which was probably due to the intertwined effect of both the yeast and gluten network [20]. When the pre-fermented time was under the optimal time, the volume of the bun was limited before freezing, and the loss of yeast activity further restricted the volume expansion. On the other hand, the excessive pre-fermented time could lead to a fragile dough structure, which is incapable of holding gas and results in a shrunken volume. The volume of par-steamed frozen dough was relatively stable during freezing and frozen storage (Figure 1C), and only the dough pre-steamed for 10 min showed a slight decrement. Bárcenas and Rosell demonstrated that the par-baked bread also had lower specific volume upon frozen storage, and suggested that the damaged gluten protein by ice crystals would not be strong enough to hold the crumb upon e further baking processes [21].

#### 3.1.2. Hardness

With an extended frozen storage time, the hardness of the steamed bun increased. The hardness of the bun with 2.0% yeast was significantly lower than that of the buns with 0.5% and 1.0% yeast during the frozen storage period (Figure 1D). By increasing the dosage of yeast, the dough could be fermented more sufficiently, resulting in a softer texture [22]. A drastic enhancement of hardness was noticed for the pre-fermented frozen dough, especially for the dough fermented for 40 min. Compared with the buns made from frozen dough pre-fermented for 20 and 40 min, frozen bun pre-fermented for 30 min had the lowest hardness (Figure 1E). Among them, the frozen dough buns with pre-steamed time of 20 min showed the lowest hardness among the three treatments when stored for 30 and 60 d (Figure 1F), whereas the hardness dramatically increased when stored for 90 and 120 d. According to Andrzej et al. [23], steamed buns with a longer pre-steamed time possessed higher free water content and induced more ice crystals during freezing, which resulted in more mechanical damage to the structure of the gluten network and further increased the hardness of the final product.

#### 3.1.3. Moisture Content

The moisture content of the skin layer as well as the inner crumb significantly affects the sensory properties of steamed buns. The crumb was the moistest, followed by the skin and the filling. The moisture of the skin layer of the bun increased slightly whereas the crumb and the filling were significantly reduced (Figure 2), suggesting that water migrated from the interior to the exterior. This phenomenon occurred because steamed bun lost water in the form of sublimation after forming ice crystals, and the moisture inside the bun continuously migrated to the outside [24]. The steamed bun with 2.0% yeast added showed stronger water holding capacity (Figure 2A). For the pre-fermented frozen dough buns (Figure 2B), pre-fermented frozen dough buns with a pre-fermented time of 30 min lost less water compared with the buns with the pre-fermented time of 20 and 40 min, which might be due to the stronger water and gluten interactions, which relieved the moisture loss [25]. For the par-steamed frozen dough bun stored for 120 d (Figure 2C), buns pre-steamed for 20 min showed the most evident water loss. Compared with the other two processes, the water content in the skin of the buns made from par-steamed frozen dough was relatively stable, which might be due to the more densely formed skin layer during the pre-steaming process, which thus held the water more tightly.

#### 3.1.4. Crumb Grain Structure

During steamed bun making, gluten proteins are cross-linked to form a three-dimensional network. The carbon dioxide produced by yeast during the fermentation process forms fine pores in the dough, conferring the buns a loose and porous structure [26]. A homogenous crumb grain structure with a higher porosity corresponds to a superior quality. The cross-section image of steamed buns made from unfermented, pre-fermented and par-steamed frozen dough are depicted in Figure 3. With the increased storage time, the porosity and average cell area decreased, whereas the cell density was enhanced (Table 2). For the unfermented frozen dough with 2.0% yeast, the optimal crumb grain structure was developed, as reflected by the higher porosity [27]. This was probably related with the stronger gassing power, thus forming a more porous structure. Compared to the unfermented frozen dough, the deterioration of the pre-fermented frozen dough was more evident. The pre-fermented time of 30 min showed the optimum structure, the pre-fermented time of 20 min was insufficient and interrupted by the freezing and the subsequent steaming process led to low porosity and smaller cells. Meanwhile, the pre-fermented time of 40 min lead to a fragile dough structure, and the dough and air bubble interface was vulnerable to damage by ice crystals, resulting in a degraded crumb structure [28]. For the par-steamed frozen dough buns stored for 120 d, the bun with a pre-steamed time of 10 and 15 min had higher porosity and lower cell density. Compared with the other two processes, the par-steamed frozen dough exhibited a more stable crumb grain structure, which was associated with the fact that the gluten proteins were polymerized during the pre-steaming process and were thus more tolerant to freezing in the par-steamed buns.

#### 3.1.5. Sensory Quality

The sensory properties indexes including color, tissue, texture, taste, odor, chewiness and stickiness were used to characterize the changes of sensory attributes of steamed buns with different processes during frozen storage (Figure 4). With the increased frozen storage time, the closed area on the radar map significantly decreased, indicating that the sensory quality of frozen dough buns with different processes decreased by varying degrees. The sensory scores of chewiness and the tissue morphology of unfermented frozen dough were significantly lowered during frozen storage (Figure 4A–C), indicating that the buns made from unfermented frozen dough were less chewable, and the skin appeared to be wrinkled or collapsed [29]. Frozen dough with 2.0% yeast possessed the largest enclosed area and thus the highest sensory quality. When frozen stored for 120 d, the 30 min pre-fermented frozen dough buns possessed a higher sensory score than the 20 and 40 min pre-fermented frozen dough buns (Figure 4D–F). Compared with the par-steamed frozen dough buns with a pre-steamed time of 20 min (Figure 4G–I), the dough pre-steamed for 10 and 15 min had a higher sensory quality during frozen storage. The amount of flavor compounds formed in steamed buns could be affected by yeast amount and activity, steaming time as well as the temperature. From the sensory quality of steamed buns, it could be concluded that the optimum processes for each frozen dough was the un-fermented frozen dough with 2% yeast, dough pre-fermented for 30 min and dough pre-steamed for 15 min, which possessed a higher sensory quality.

#### 3.1.6. Gassing Power

During the frozen storage period, the gas production of yeast in the un- and pre-fermented frozen dough significantly decreased (Figure 5). The loss of yeast viability and activity in the unfermented frozen dough buns led to decreased gas production [30]. Increasing the amount of yeast could increase the survival of yeast and thus of the gas production. Under the fixed frozen storage time, the gas production of the pre-fermented frozen dough was lower than that of the unfermented frozen dough, with less gassing power detected for the longer pre-fermented period. After being activated upon fermentation, the activated metabolism of yeast could induce more free water in the cell, and thus form more ice crystals. Thus, the membrane of yeast was physically disrupted by ice recrystallization as well as the intracellular and extracellular osmotic pressure difference, which further resulted in cellular damage and led to decreased gas production [5].

### 3.2. The Influence of Different Processing Techniques on the Gluten Structure

The steamed buns made from the unfermented frozen dough with 2% yeast, dough pre-fermented for 30 min and dough par-steamed for 15 min showed the best sensory quality. Therefore, the free sulfhydryl (SH) content, molecular weight distribution and surface hydrophobicity of the gluten protein of frozen dough and corresponding steamed buns were analyzed to illustrate the gluten structure changes, which could explain the different sensory quality of steamed buns.

#### 3.2.1. Free SH Content

The free SH contents were determined to characterize the state of disulfide (SS) bonds of gluten protein (Table 3). With the increased frozen storage time, the SH of all the frozen dough increased significantly, indicating that frozen storage induced the fracture of SS bonds and the depolymerization of gluten proteins. Ribotta et al. [31] suggested that water redistribution and recrystallization of ice are the main reasons for the breakage of SS bonds. For the un- and pre-fermented dough, the glutathione (GSH) released by the dead yeast could further affect the dough by breaking SS bonds of the viscoelastic gluten network [32]. For the par-steamed frozen dough buns, the SH content was much lower than those of the un- and pre-fermented buns. During the pre-steaming process of par-steamed buns, SH participated in the polymerization of glutenin and gliadin through the oxidation of SH and SH/SS exchange reaction [33], resulting in the reduction of SH levels. The contents of SH in pre-fermented frozen dough buns were significantly higher than those of the other two processes, indicating that SS bonds in gluten protein were more likely to break after pre-fermentation. After steaming, the SH content of buns made from frozen dough buns were significantly reduced compared with the ones made from fresh dough, which was due to the suppressed polymerization of glutenin and gliadin induced by frozen storage.

#### 3.2.2. Molecular Weight Distribution of Gluten Protein

The solubility in SDS solution is an indicator of the covalent crosslinking degree of protein, as SDS could disrupt non-covalent interactions [34]. As shown in Figure 6A, two major fractions of gluten in the unheated dough can be distinguished by size exclusion (SE)-HPLC profile: SDS-soluble polymers (SDS-P), with the Mw ranging from 88,000 to 698,000, mainly constituted by high molecular weight glutenin polymers; SDS-soluble monomers (SDS-M), with the Mw ranging from 28,000 to 88,000, consisting of monomeric gliadin and glutenin fractions and salt-soluble proteins with Mw below 28,000. The Mw distribution of the SDS-soluble gluten of steamed buns remained constant, whereas the content of each fraction diminished significantly (Figure 6B), which was attributed to the extensive heat-induced polymerization of glutenin and gliadin upon steaming.

For the unheated dough, the SDS-I were designated as glutenin marcropolymers (GMP), whereas the drastically enhanced SDS-I in the par-steamed bun was caused by the formation of glutenin-gliadin macro-crosslinking. For all the frozen dough, the SDS-P and SDS-M content significantly increased at the expense of SDS-I with extended storage time (Table 4), indicating that depolymerization of glutenin macropolymers occurred. After 120 d of frozen storage, depolymerization by 67.82%, 45.03% and 22.08% were detected for the unfermented, pre-fermented and par-steamed frozen doughs, respectively. The superior freezing tolerance of par-steamed dough was probably related to stable glutenin–gliadin macro-crosslinking. After steaming, the intensive decrease of SDS-P and SDS-M suggested that polymerization of glutenin and gliadin formed massive SDS-insoluble glutenin–gliadin crosslinking upon steaming. For the steamed bread made from un- or pre-fermented frozen dough, the reduced SDS-I level in frozen dough was associated with the weakened polymerization ability of glutenin and gliadin [35]. In addition to the water content loss upon frozen storage, the total SDS-soluble proteins were also suggested to be positively correlated with the firmness of steamed bread [9]. For the par-steamed frozen dough, the variation of SDS-I content upon further steaming was less evident as compared with the other two types. This was probably due to the already established polymerized gluten network in the par-steamed dough, which was also relatively stable upon steaming.

#### 3.2.3. The Effect of Different Processes on the Surface Hydrophobicity of Gluten Protein

Surface hydrophobicity (H_0_) reflects the number of hydrophobic groups on the surface of gluten protein, which can indicate the tertiary structure of proteins [16]. With prolonged frozen storage, H_0_ of the gluten protein for all the frozen dough steadily decreased (Figure 6C), which indicated surface hydrophobic groups being buried. Freezing-evoked protein denaturation was primarily attributed to the elevated contacted interface with the ice layer, resulting in the surface-induced denaturation of proteins [36]. The H_0_ of par-steamed dough was significantly lower than the other two doughs, which was caused by the inter-molecular aggregation of proteins via the SS bridge and hydrophobic interactions. This could induce the folding of proteins and bury the hydrophobic groups [37]. Moreover, the H_0_ of gluten proteins of frozen dough steamed buns decreased during the initial storage and were subsequently enhanced (Figure 6D), which was related with the combined effects of the unfolding and synchronous aggregation induced by freezing and steaming, respectively. Moreover, H_0_ of par-steamed frozen dough and steamed buns were more resistant to frozen storage, which was probably due to the established stable aggregated structure during the pre-steaming process.

## 4. Conclusions

During the frozen storage period, steamed buns made from unfermented dough with 2.0% yeast addition, the pre-fermented dough with a pre-fermented time of 30 min and the par-steamed dough with pre-steamed time of 15 min showed the best sensory properties. During the frozen storage period, the gluten proteins in the par-steamed buns were the most freeze-tolerant, whereas the freezing-induced depolymerization of gluten proteins was most distinct in the unfermented dough, which was probably related to the superior freeze-stability of glutenin–gliadin macro-crosslinks. For the par-steamed frozen dough, the variation of SDS-I content upon further steaming was less evident as compared with the other two types. The surface hydrophobicity of gluten proteins in the frozen dough steamed buns decreased during the initial storage and were enhanced subsequently, which was associated with the combined effects of the unfolding and the further aggregation induced by freezing and steaming, respectively.

## Figures and Tables

**Figure 1 foods-11-03833-f001:**
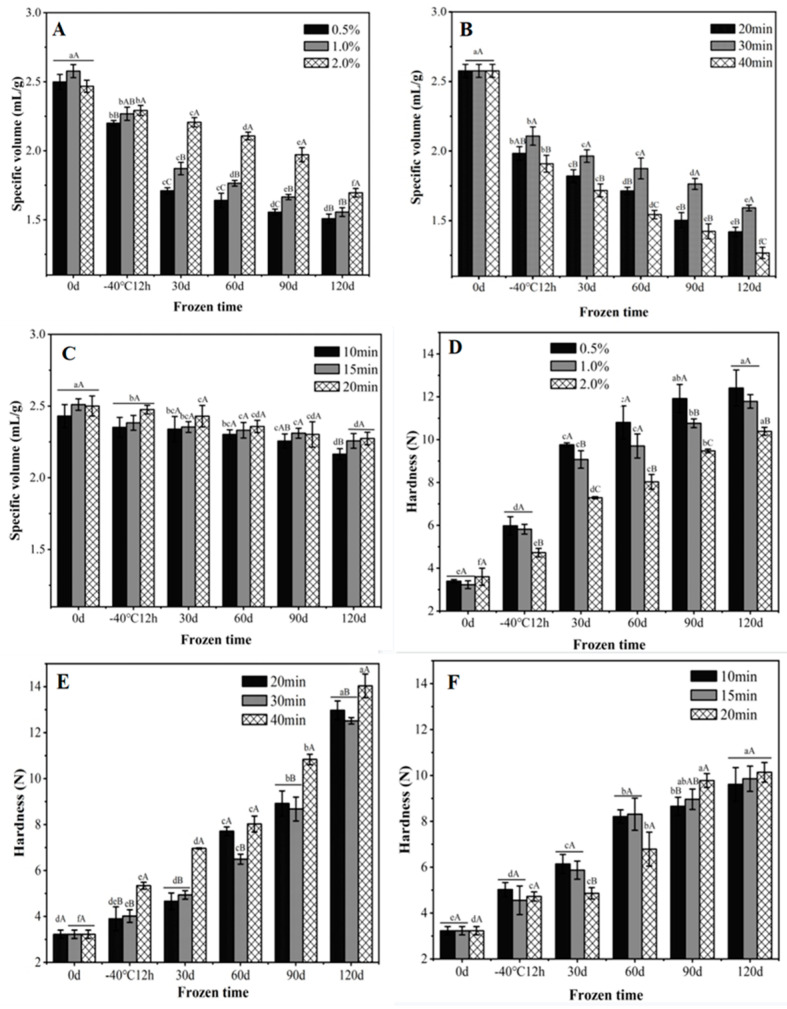
Effects of frozen storage on the specific volume and hardness of steamed buns made from unfermented frozen dough with yeast additions of 0.5%, 1.0% and 2.0% (**A**,**D**); pre-fermented frozen dough with the pre-fermented time of 20, 30 and 40 min (**B**,**E**); the par-steamed frozen dough with the pre-steamed time of 10, 15 and 20 min (**C**,**F**). Data with different lowercase and uppercase letters indicate the significant differences among the frozen dough with different storage time and different processing variables, respectively (*p* < 0.05).

**Figure 2 foods-11-03833-f002:**
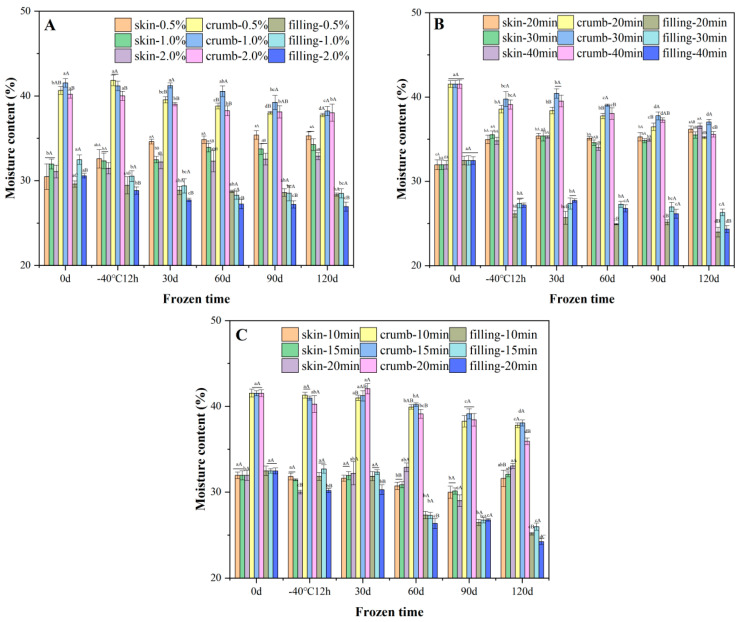
Effects of frozen storage on the moisture content distribution of steamed buns made from unfermented (**A**), pre-fermented (**B**) and par-steamed (**C**) dough. Data with different lowercase and uppercase letters indicate the significant differences among the frozen doughs with different storage times and different processing variables, respectively (*p* < 0.05).

**Figure 3 foods-11-03833-f003:**
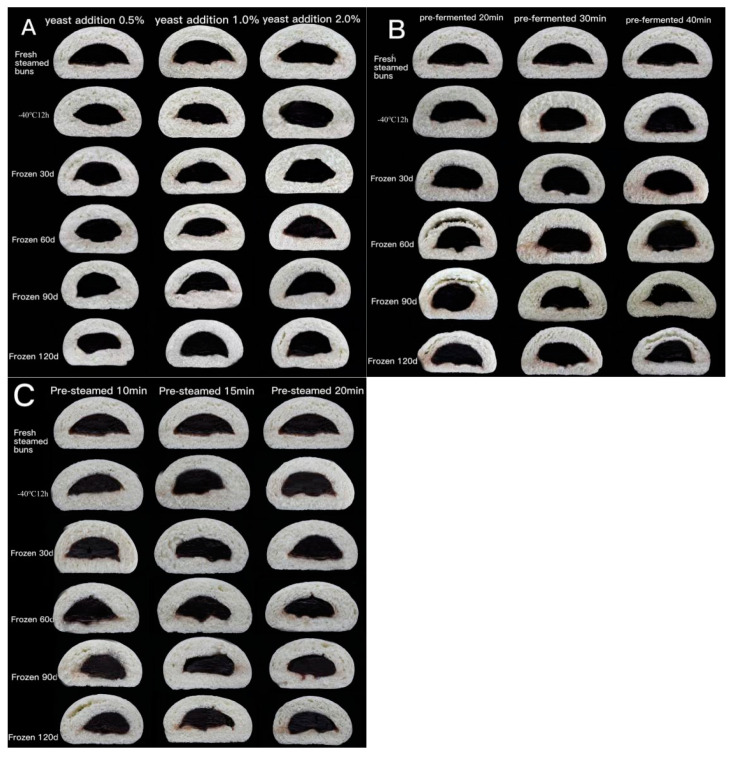
The cross section image of steamed buns made from un-fermented (**A**), pre-fermented (**B**) and par-steamed (**C**) frozen dough.

**Figure 4 foods-11-03833-f004:**
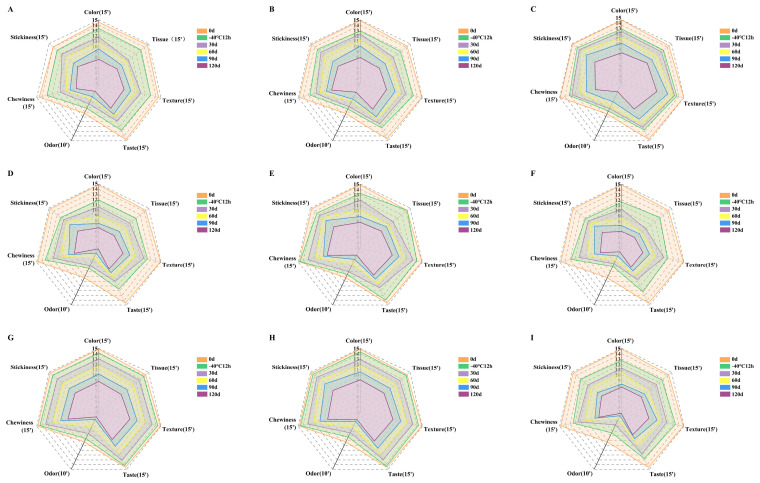
The sensory evaluation score of steamed bum made from un-fermented frozen dough with yeast dosage of 0.5% (**A**), 1.0% (**B**) and 2.0% (**C**), the pre-fermented time of 20 (**D**), 30 (**E**) and 40 min (**F**), and pre-steamed time of 10 (**G**), 15 (**H**) and 20 min (**I**). The scores in the brackets are the maximum scores for each sensory attribute.

**Figure 5 foods-11-03833-f005:**
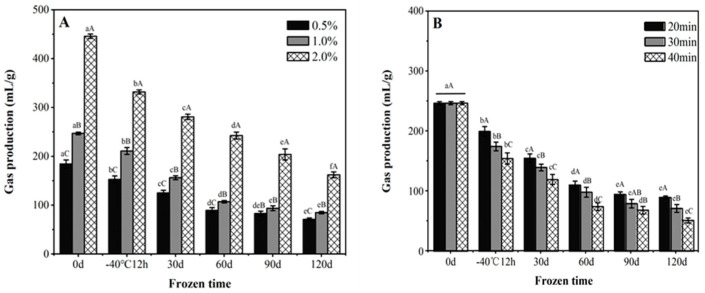
The effect of frozen storage on the gas production capacity of unfermented (**A**) and pre-fermented (**B**) frozen dough. Data with different lowercase and uppercase letters indicate the significant differences among the frozen doughs with different storage times and different processing variables, respectively (*p* < 0.05).

**Figure 6 foods-11-03833-f006:**
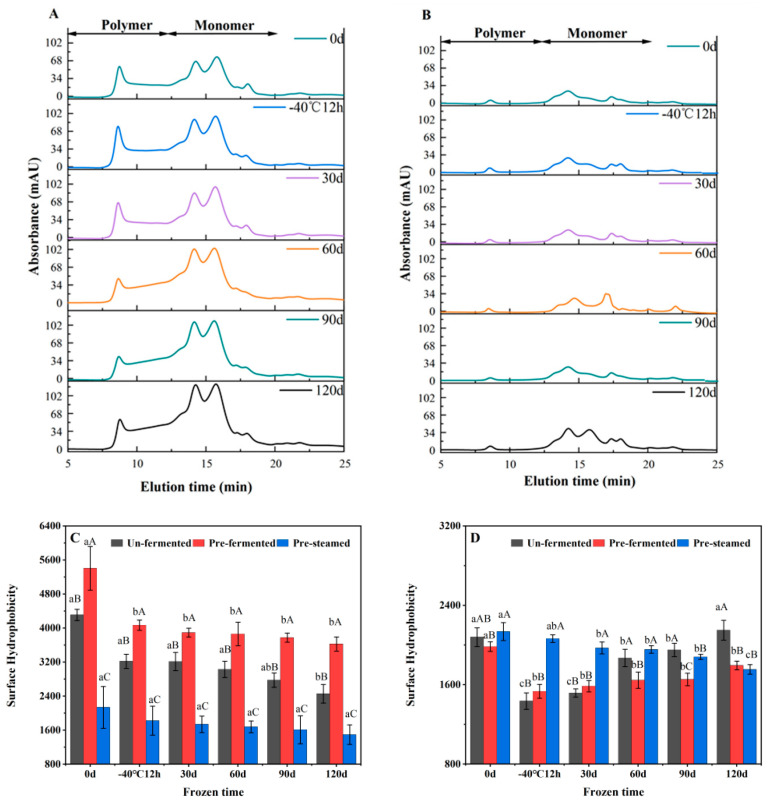
Representative SE-HPLC profile of unfermented (**A**) and par-steamed frozen dough buns(**B**), and the variation of surface hydrophobicity of gluten proteins in doughs (**C**) and steamed buns (**D**) during frozen storage. Data with different lowercase and uppercase letters indicate the significant differences among the frozen doughs with different storage times and different processing techniques, respectively (*p* < 0.05).

**Table 1 foods-11-03833-t001:** Scoring method for sensory evaluation of steamed buns.

Parameters	Score	Evaluation Rules
Color	15	White/creamy white (11–15), little yellow (6–10), gray, or dark (1–5)
Tissue	15	Very smooth, bright, no specks (11–15), rough surface, shrinking skin, specks or bubbles in skin (6–10), skin damage (1–5)
Texture	15	Good resilience when pressed with finger and bite a little hard stress (11–15), rebounds slowly and bite with a little stress (6–10), and poor resilience and crumbly (1–5)
Taste	15	Sweet, filling aroma obvious (8–10), the taste is flat and the aroma is not outstanding (4–7), and musty or abnormally smell poor (1–3)
Odor	10	A pleasant smell, no peculiar smell (8–10), smells flat (4–7), and poor very unpleasant (1–3)
Chewiness	15	Not rough (11–15), a little rough (6–10), and very rough (1–5)
Stickiness	15	Not sticky (11–15), a little sticky (6–10), and very sticky (1–5)

**Table 2 foods-11-03833-t002:** Effects of different processes on the internal pore structure parameters of frozen dough buns.

	Frozen Time	Unfermented Frozen Dough Buns			Pre-Fermented Frozen Dough Buns			Par-Steamed Frozen Dough Buns		
		0.5%	1.0%	2.0%	20 min	30 min	40 min	10 min	15 min	20 min
Porosity(%)	0 d	54.35 ± 2.50 ^aAB^	57.86 ± 2.41 ^aA^	53.88 ± 0.80 ^aB^	54.49 ± 1.09 ^aA^	55.03 ± 1.03 ^aA^	57.86 ± 2.41 ^aA^	57.86 ± 2.41 ^aA^	57.86 ± 2.41 ^aA^	57.86 ± 2.41 ^aA^
	−40 °C 12 h	50.46 ± 1.19 ^aB^	54.74 ± 1.15 ^aA^	55.95 ± 1.47 ^aA^	48.89 ± 0.52 ^bB^	53.17 ± 2.36 ^abA^	46.08 ± 0.94 ^bC^	56.52 ± 1.05 ^aA^	57.39 ± 1.19 ^aA^	55.89 ± 2.94 ^aA^
	30 d	46.30 ± 1.59 ^bB^	48.54 ± 0.79 ^bB^	53.54 ± 0.81 ^aA^	42.77 ± 1.83 ^cB^	48.87 ± 3.42 ^bA^	39.04 ± 3.89 ^cB^	54.05 ± 3.74 ^abA^	55.87 ± 3.10 ^aA^	53.42 ± 2.22 ^aA^
	60 d	40.12 ± 2.38 ^cB^	43.41 ± 1.10 ^cB^	49.27 ± 2.97 ^bA^	38.64 ± 1.35 ^dB^	44.35 ± 0.35 ^cA^	35.01 ± 0.40 ^cdC^	50.73 ± 3.39 ^bA^	49.91 ± 2.38 ^bA^	47.33 ± 2.71 ^bA^
	90 d	33.51 ± 1.61 ^dC^	37.27 ± 0.40 ^dB^	43.45 ± 1.13 ^cA^	32.72 ± 0.92 ^eB^	38.85 ± 2.42 ^dA^	31.09 ± 3.71 ^dB^	45.23 ± 1.38 ^cA^	46.74 ± 3.24 ^bA^	43.68 ± 3.19 ^bA^
	120 d	30.18 ± 0.64 ^eB^	32.02 ± 0.29 ^eB^	36.97 ± 1.46 ^dA^	28.52 ± 2.78 ^fB^	34.28 ± 1.24 ^eA^	25.55 ± 2.88 ^eB^	38.60 ± 0.80 ^dB^	40.67 ± 0.26 ^cA^	37.42 ± 0.20 ^cB^
Cell density (cells/cm^2^)	0 d	42.78 ± 3.16 ^eA^	43.54 ± 3.70 ^eA^	43.89 ± 2.62 ^dA^	42.41 ± 2.87 ^eA^	43.25 ± 3.85 ^dA^	43.54 ± 3.70 ^eA^	43.54 ± 3.70 ^eA^	43.54 ± 3.70 ^dA^	43.54 ± 3.70 ^eA^
	−40 °C 12 h	55.04 ± 4.06 ^dA^	52.53 ± 2.08 ^dA^	51.65 ± 4.31 ^cA^	58.04 ± 3.38 ^dAB^	55.02 ± 1.85 ^cB^	61.47 ± 2.15 ^dA^	46.88 ± 1.25 ^deA^	47.82 ± 5.01 ^cdA^	49.78 ± 2.66 ^dA^
	30 d	60.14 ± 2.13 ^cdA^	56.32 ± 5.78 ^cdA^	54.73 ± 4.55 ^cA^	63.04 ± 5.70 ^cdAB^	58.73 ± 2.93 ^cB^	67.40 ± 4.10 ^cA^	52.16 ± 2.55 ^cdA^	52.54 ± 1.63 ^cA^	54.13 ± 3.29 ^cdA^
	60 d	67.93 ± 7.68 ^bcA^	62.86 ± 3.76 ^bcA^	58.98 ± 5.87 ^bcA^	69.82 ± 2.01 ^cA^	65.99 ± 4.52 ^bA^	72.11 ± 4.77 ^cA^	56.33 ± 2.82 ^bcA^	57.10 ± 0.61 ^bA^	58.41 ± 3.70 ^bcA^
	90 d	75.25 ± 8.85 ^abA^	67.33 ± 4.93 ^bA^	63.21 ± 4.98 ^abA^	77.23 ± 1.79 ^bB^	70.86 ± 2.95 ^abC^	81.95 ± 1.32 ^bA^	60.75 ± 5.27 ^abA^	60.11 ± 5.57 ^abA^	62.71 ± 1.16 ^abA^
	120 d	80.52 ± 3.30 ^aA^	75.56 ± 2.07 ^aA^	69.59 ± 2.15 ^aB^	84.30 ± 2.35 ^aA^	76.18 ± 5.40 ^aB^	88.11 ± 1.77 ^aA^	65.71 ± 2.44 ^aA^	64.49 ± 5.30 ^aA^	67.25 ± 4.53 ^aA^
Cell average area (mm^2^)	0 d	1.27 ± 0.08 ^aA^	1.33 ± 0.03 ^aA^	1.32 ± 0.08 ^aA^	1.28 ± 0.02 ^aA^	1.29 ± 0.10 ^aA^	1.33 ± 0.03 ^aA^	1.33 ± 0.03 ^aA^	1.33 ± 0.03 ^aA^	1.33 ± 0.03 ^aA^
	−40 °C 12 h	0.92 ± 0.03 ^bB^	1.04 ± 0.01 ^bA^	1.08 ± 0.05 ^bA^	0.84 ± 0.06 ^bB^	0.97 ± 0.09 ^bA^	0.75 ± 0.08 ^bB^	1.16 ± 0.07 ^bA^	1.20 ± 0.04 ^bA^	1.12 ± 0.07 ^bA^
	30 d	0.77 ± 0.07 ^cB^	0.86 ± 0.10 ^cAB^	0.98 ± 0.10 ^bcA^	0.68 ± 0.01 ^cB^	0.83 ± 0.02 ^cA^	0.58 ± 0.03 ^cC^	1.04 ± 0.05 ^cA^	1.06 ± 0.07 ^cA^	1.07 ± 0.01 ^bA^
	60 d	0.59 ± 0.07 ^dB^	0.69 ± 0.05 ^dB^	0.84 ± 0.04 ^cA^	0.55 ± 0.03 ^dB^	0.67 ± 0.07 ^dA^	0.49 ± 0.06 ^cB^	0.90 ± 0.08 ^dA^	0.87 ± 0.01 ^dA^	0.81 ± 0.08 ^cA^
	90 d	0.45 ± 0.04 ^eB^	0.55 ± 0.08 ^eB^	0.69 ± 0.02 ^dA^	0.42 ± 0.05 ^eB^	0.55 ± 0.04 ^deA^	0.38 ± 0.04 ^cdB^	0.74 ± 0.06 ^eA^	0.73 ± 0.10 ^eA^	0.70 ± 0.10 ^cA^
	120 d	0.37 ± 0.01 ^fB^	0.42 ± 0.09 ^fA^	0.53 ± 0.02 ^eA^	0.34 ± 0.03 ^eB^	0.45 ± 0.05 ^eA^	0.29 ± 0.07 ^dB^	0.59 ± 0.03 ^fA^	0.63 ± 0.07 ^eA^	0.56 ± 0.04 ^dA^

Data with different lowercase letters in the same column and capital letters in the same row for the same type of frozen dough indicate a significant difference (*p* < 0.05).

**Table 3 foods-11-03833-t003:** Effects of frozen storage time on free SH content of thawed dough and steamed buns.

Free SH Content (μmol/g)						
Frozen Time	Thawed Dough			Steamed Bun		
	Unfermented	Pre-Fermented	Par-Steamed	Unfermented	Pre-Fermented	Par-Steamed
0 d	1.11 ± 0.01 ^fA^	1.05 ± 0.02 ^fA^	0.46 ± 0.04 ^fB^	0.87 ± 0.04 ^eA^	0.60 ± 0.06 ^fB^	0.46 ± 0.04 ^eC^
−40 °C 12 h	1.32 ± 0.03 ^eB^	1.49 ± 0.03 ^eA^	0.59 ± 0.02 ^eC^	0.92 ± 0.03 ^deB^	1.05 ± 0.02 ^eA^	0.50 ± 0.02 ^eC^
30 d	1.57 ± 0.05 ^dA^	1.61 ± 0.04 ^dA^	0.81 ± 0.02 ^dB^	1.00 ± 0.03 ^dB^	1.16 ± 0.03 ^dA^	0.59 ± 0.03 ^dC^
60 d	1.84 ± 0.10 ^cB^	2.25 ± 0.04 ^cA^	1.13 ± 0.06 ^cC^	1.36 ± 0.06 ^cB^	1.46 ± 0.01 ^cA^	0.90 ± 0.05 ^cC^
90 d	2.35 ± 0.05 ^bB^	2.58 ± 0.06 ^bA^	1.34 ± 0.04 ^bC^	1.59 ± 0.05 ^bB^	1.74 ± 0.05 ^bA^	1.26 ± 0.01 ^bC^
120 d	2.55 ± 0.08 ^aB^	3.20 ± 0.10 ^aA^	1.67 ± 0.04 ^aC^	1.79 ± 0.03 ^aB^	1.93 ± 0.02 ^aA^	1.36 ± 0.03 ^aC^

Data with different lowercase letters in the same column and capital letters in the same row for dough or bun indicate a significant difference (*p* < 0.05).

**Table 4 foods-11-03833-t004:** Effects of frozen storage time on protein molecular weight distribution after thawing and steaming of frozen dough buns with different processes.

Protein Molecular Weight Distribution (%)							
	Frozen Time	Thawed Dough			Steamed Bun		
		Unfermented	Pre-Fermented	Par-Steamed	Unfermented	Pre-Fermented	Par-Steamed
SDS-P	0 d	9.42 ± 0.47 ^fA^	2.97 ± 0.31 ^eB^	1.06 ± 0.08 ^dC^	0.81 ± 0.11 ^eA^	0.82 ± 0.15 ^dA^	0.77 ± 0.08 ^dA^
	−40 °C 12 h	10.18 ± 0.32 ^eA^	3.44 ± 0.29 ^eB^	1.12 ± 0.06 ^cdC^	1.02 ± 0.08 ^deA^	0.99 ± 0.19 ^cdA^	0.82 ± 0.17 ^dA^
	30 d	10.97 ± 0.26 ^dA^	4.36 ± 0.16 ^dB^	1.34 ± 0.04 ^bcC^	1.20 ± 0.11 ^dA^	1.17 ± 0.07 ^bcA^	1.29 ± 0.05 ^cA^
	60 d	13.17 ± 0.61 ^cA^	6.40 ± 0.41 ^cB^	1.46 ± 0.09 ^bC^	1.67 ± 0.23 ^cA^	1.35 ± 0.03 ^abA^	1.41 ± 0.11 ^bcA^
	90 d	16.43 ± 0.96 ^bA^	7.58 ± 0.32 ^bB^	1.61 ± 0.24 ^abC^	2.21 ± 0.16 ^bA^	1.46 ± 0.07 ^aB^	1.57 ± 0.07 ^abB^
	120 d	18.28 ± 0.53 ^aA^	8.98 ± 0.19 ^aB^	1.86 ± 0.09 ^aC^	2.68 ± 0.19 ^aA^	1.58 ± 0.11 ^aB^	1.73 ± 0.12 ^aB^
SDS-M	0 d	64.32 ± 0.80 ^cA^	63.5 ± 0.85 ^cA^	17.52 ± 0.44 ^cB^	14.97 ± 0.35 ^eB^	27.21 ± 0.87 ^dA^	15.83 ± 0.44 ^dB^
	−40 °C 12 h	65.91 ± 0.96 ^bcA^	64.58 ± 0.97 ^cA^	23.19 ± 0.87 ^cB^	17.03 ± 0.33 ^dB^	28.69 ± 0.76 ^dA^	17.97 ± 0.79 ^cB^
	30 d	66.82 ± 2.33 ^bcA^	66.84 ± 0.84 ^bA^	27.62 ± 0.86 ^bB^	18.19 ± 0.41 ^cC^	30.69 ± 0.76 ^cA^	21.01 ± 0.77 ^bB^
	60 d	68.61 ± 3.12 ^abA^	68.97 ± 2.54 ^bA^	29.39 ± 1.25 ^bB^	20.52 ± 0.18 ^bC^	32.55 ± 0.20 ^bA^	22.37 ± 0.84 ^bB^
	120 d	73.27 ± 2.89 ^aA^	72.59 ± 0.70 ^aA^	34.70 ± 0.66 ^aB^	22.20 ± 1.18 ^aC^	35.89 ± 0.68 ^aA^	29.46 ± 0.74 ^aB^
SDS-I	0 d	26.26 ± 0.39 ^aC^	33.53 ± 0.96 ^aB^	81.42 ± 0.30 ^aA^	84.22 ± 0.33 ^aA^	71.97 ± 0.76 ^aB^	83.41 ± 0.51 ^aA^
	−40 °C 12 h	23.91 ± 0.78 ^bC^	31.98 ± 0.82 ^abB^	75.69 ± 0.91 ^bA^	81.95 ± 0.31 ^bA^	70.32 ± 0.92 ^aB^	81.21 ± 0.81 ^bA^
	30 d	22.22 ± 2.10 ^bcC^	28.80 ± 0.99 ^bB^	71.04 ± 0.86 ^bA^	80.61 ± 0.43 ^cA^	68.14 ± 0.74 ^bC^	77.70 ± 0.77 ^cB^
	60 d	18.22 ± 2.90 ^cC^	24.63 ± 2.92 ^cB^	69.16 ± 1.30 ^bA^	77.81 ± 0.05 ^dA^	66.10 ± 0.17 ^cC^	76.22 ± 0.95 ^cB^
	90 d	13.57 ± 1.94 ^dC^	22.73 ± 2.15 ^cB^	65.09 ± 1.08 ^cA^	75.98 ± 0.21 ^eA^	64.78 ± 0.94 ^cC^	73.33 ± 0.76 ^dB^

Data with different lowercase letters in the same column and capital letters in the same row for dough or bun indicate a significant difference (*p* < 0.05).

## Data Availability

Data is contained within the article.

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
