# Peer review of "Impact of Different Frozen Dough Technology on the Quality and Gluten Structure of Steamed Buns"

_foods, 2022, doi:10.3390/foods11233833_

Round 1

Reviewer 1 Report

The methodologies were well designed to obtain appropriate results. Analysis of bread quality (gasification and density) was correlated with texture data. The relationship of these results was explained by chemical techniques. It was a very successful and relevant work for the bakery industry.

The work could be completed with rheological tests such as creep and recovery tests, viscoelasticity tests.

Reviewer 2 Report

Line 41: “ … custard[1].In China’ – add the space before the parentheses and after the dot – check the whole manuscript (for example lines 59, 119, 130, 145, etc.)

 Line 63: „.. bun. [6] evaluated..” – It looks as if something is missing in this sentence – please rewrite. Who evaluated? “Wang et al. evaluated … [6].”

 Materials and methods

Was the control sample established for this study? If yes please indicate.

Line 77-76: “The results…” – This sentence looks more like a conclusion. In this part of the manuscript, you should write the study's aim, not the conclusion. I propose a little correction for example  “The results of this study could contribute …

Line 85: “experiment., and” – delete the dot

Line 94: “..the steamed bun dough..” - I propose to delete the word “steamed”, it can suggest that the dough was steamed before the frozen process

Line 100: bread? It should be the bun. – In some parts of the manuscript you use the word bread instead of a bun, please check and correct (some places are listed below)

Line 104 and 107: was the same

Line 112:  probe. [10] – the dot should be placed after the parentheses

Line 114: “The bread slices..” – It is not clear

Line 115: “The central part of steamed bread (2 g) was taken for 115 moisture analysis” – It should be a bun, not bread. Moreover, the central part of the bun is red bean paste filling, so the moisture refers to the dough or the filling? In line 207 there is a suggestion that the moisture of the skin, inner crumb, and also filling was measured.

Line 117: “bread images”??? –

Line 120: “was performed by twenty trained panelists (15 females and 15 males.” – There is a mistake the total number of panelists is 30, not 20

Line 121: please unify the name of sensory features so that the same name should be used in the Methods section, Figure 4, and discussion.

Line 236: “optimal crumb grain structure” – what do you mean by optimal structure, please define the features of such structure.

 Results and Discussion

Line 179:optimum specific volume.” – What do you exactly mean by optimum volume, what were the criteria of optimization? You can say “the highest, the lowest, etc.” but when you are using the word “optimum” it means that there was some kind of optimization criteria that you should establish before the experiment (not always the highest means optimum).

Line 186: “[20] demonstrated …”? – Who demonstrated? It would be better to write “Bárcenas and Rosell demonstrated …. [20].”

 Line 257: viscosity? Viscosity refers to fluid. How the viscosity of buns (solid) was measured?

Line 361: It should be Figure 6D

 Conclusion

Line 383: “the optimal organoleptic quality”? What do you mean by optimal organoleptic quality? Please specify the optimal values for each organoleptic feature. Indicate the target group for which these are the optimal quality features.

 Figures and Tables

Tables 1, 2, and 3 - You use lower and capital letters to show the statistical differences between samples, but the description below the table did not explain the purpose of using lower and capital letters.

Figure 1 and 2 and tables – What “-40oC12h” means, that appears in the x-axis description or tables? The time for freezing given in the Methods sections is 24 h.

 Also please add additional descriptions for example legends in Figure 1 (“0.5%. 1%. .. 10 min, 20 min etc). The figure and tables should be self-explanatory so that a reader can determine what the figure or table is showing without having to look for additional information in the text of the article.

Reviewer 3 Report

The word organoleptic is in disuse. Nowadays the sensory attribute measured is specified, or "sensory properties" may be mentioned.
It needs to be made clear how sensory properties were evaluated, is intensity measured, or is it a hedonic scale?
How an optimal sensory quality is defined.
In the comparison of means, why was Duncan used instead of Tukey?
In the tables, why is a comparison of means made in the columns?
In the discussion of results, more references should have been used.
It is suggested to use mol/L instead of Molarity. What is mM/L?

Line 22, 49, 256 and 383  - organoleptic

Line 42. …“Custar[1] .In China” correct to “Custar [1]. In China”…

Line 116 mM/L ???

Line 122 and 128 – CO2 correct to CO2

Line 133 – 800g to 800 g

139 mM/L??

Line 145 – “) . [17]The” to “) [17]. The” – Review the way citations are written throughout the text, whether they are placed before or after the period.

Round 2

Reviewer 3 Report

The references presented for the sensory evaluation methodology use a 9-point hedonic scale (lines 118-120). Radhiah Shukri agrees with the sensory evaluation methodology, and they use untrained consumers. However, Li uses trained judges, as he rates acceptance and intensity.
You can change the number of points on the scale. However, usually, a 9-point hedonic scale is used.
On the other hand, radar charts (line 200) have two scales, a 15-point, and a 10-point scale, and at least to the naked eye, they look balanced (15 equals 10).
The discussion of the sensory evaluation results is based on the intensity of the attributes when the methodology specified a hedonic scale. In the conclusions, optimal sensory properties continue to be used.

Line 409: LWT - Food Sci. Technol.2009. Change to LWT - Food Sci. Technol. 2009.

Revise the format of the references. The suggested format should be used.

Journal Articles:
1. Author 1, A.B.; Author 2, C.D. Title of the article. Abbreviated Journal Name Year, Volume, page range.
